# Seismic reflections from a lithospheric suture zone below the Archaean Yilgarn Craton

Andrew J. Calvert [1✉], Michael P. Doublier [2,3] & Samantha E. Sellars [1]

Seismic reflectors in the uppermost mantle, which can indicate past plate tectonic subduction, are exceedingly rare below Archaean cratons, and restricted to the Neoarchaean. Here we present reprocessed seismic reflection profiles from the northwest Archaean Yilgarn Craton and the Palaeoproterozoic Capricorn Orogen of western Australia that reveal the existence of a ~4 km thick south-dipping band of seismic reflectors that extends from the base of the Archaean crust to at least 60 km depth. We interpret these reflectors, which lie south of a ~50 km deep crustal root, as a relict suture zone within the lithosphere. We suggest that the mantle reflectors were created either by subduction of an oceanic plate along the northern edge of the Yilgarn Craton, which started in the Mesoarchaean and produced the rocks in northern Yilgarn greenstone belts that formed in a supra-subduction zone setting, or, alternatively, by underthrusting of continental crust deep into the lithosphere during the Palaeoproterozoic.

[1] Department of Earth Sciences, Simon Fraser University, Burnaby, British Columbia V5A 1S6, Canada. [2] Mineral Systems Branch, Geoscience Australia, Symonston, ACT 2609, Australia. [3] Centre for Exploration Targeting, School of Earth and Environment, The University of Western Australia, 35 Stirling Highway, Crawley, WA 6009, Australia. ✉email: acalvert@sfu.ca

Earth's tectonic regime has evolved as the planet cooled with the Archaean eon likely marking the change from a hot stagnant lid[1–3] or plutonic squishy lid[4] tectonic regime with intense plume-related magmatism and possible mantle overturn[5] to an early form of plate tectonics with subduction that may initially have been episodic. Numerical thermomechanical modelling has explored the effects of different thermal regimes on plate rheology and buoyancy when a mafic oceanic plate converges with continental crust, and identified three classes of response to the horizontal displacement as mantle temperatures decrease: (1) nonsubduction where weak plates accommodate horizontal movement by internal strain, (2) presubduction where convergence causes shallow underthrusting of the oceanic plate, and (3) one-sided subduction where the oceanic plate descends into the mantle[6]. On Earth, the transition to an early form of plate subduction likely occurred during the Meso-to Neoarchaean when mantle temperatures were 150–250 °C hotter than present-day values, though slab-break off could have been more common causing subduction to be shorter-lived and more intermittent than today[7].

Observations of dipping seismic reflections in the uppermost mantle below Archaean cratons can provide important constraints on the timing of the initiation of plate subduction on Earth, and potentially discriminate between the plate subduction and presubduction tectonic regimes in which higher mantle temperatures may result in low-angle underthrusting, buckling and imbrication of an oceanic plate. During present-day subduction, the boundary between the descending oceanic plate and the overriding plate can commonly be imaged as a zone of seismic reflectors up to 6 km thick that dips towards the volcanic arc and extends to depths of 60 km or more. Similar zones of dipping reflections have been observed in the uppermost mantle below Phanerozoic, Proterozoic and Archaean collision zones, where these reflectors are interpreted to be relict scars created by plate subduction[8–11]. The clearest example of such mantle reflections below an Archaean craton is located beneath the Opatica plutonic belt of the eastern Superior Craton, and formed during the rapid southward growth of the craton in the Neoarchaean at approximately 2.69 Ga[12]; however, shorter sets of reflectors that extend up to 10 km into the uppermost mantle near offsets in the Moho have been interpreted as due to subduction at 2.69–2.68 Ga and 2.71–2.70 Ga in the western Superior Craton[13] and at 2.65–2.58 Ga in the Slave Craton[14,15]. To date, similar mantle reflections linked to deformation in the overlying crust have not been observed below other Archaean cratons, including the Yilgarn Craton of western Australia.

The core of the Yilgarn Craton[16–18] is the ~3.05–2.60 Ga Youanmi Terrane, which is separated by the Ida Fault from a series of >2.95–2.65 Ga terranes to the east that were either accreted and reworked during the Neoarchaean[19] or developed through interaction with a mantle plume[20], and now form the Eastern Goldfields Superterrane. The Youanmi Terrane contains NW to NE striking greenstone belts surrounded by granite and granitic gneiss. The final phase of crustal assembly occurred during a prolonged period of intermittent E-W shortening from >2.73–2.65 Ga[21]. This shortening, which probably involved orogen-parallel escape, may be related to accretion of the 3.7–3.0 Ga Narryer Terrane over the northwestern margin of the Youanmi Terrane and amalgamation with the Eastern Goldfields Superterrane[18,22–24]. The identification of boninites near the bases of both the 2.82–2.80 Ga Norie and 2.80–2.74 Ga Polelle groups of the Meekatharra-Cue greenstone belt in the northwest Yilgarn Craton (Fig. 1), led to the proposal that a subduction zone was active from 2.82–2.74 Ga along the northwest margin of the Yilgarn Craton[25,26], and drove accretion of the Narryer Terrane[22,27], which was followed by intrusion of granites into

both terranes at 2.66 Ga[28]. Cratonization concluded with a late granite bloom consisting of high-temperature crustal melts from 2.66–2.61 Ga, which coincided with most of the gold mineralisation[19,29,30].

The <2.555 Ga Glenburgh Terrane, which contains inherited zircons as old as 3.447 Ga, formed separately from the Pilbara and Yilgarn cratons[31]. The terrane was accreted to the southern edge of the Pilbara Craton during the Opthalmian Orogeny from 2.215–2.145 Ga[32–34], and a continental arc subsequently developed along the southern margin of this composite terrane, as evidenced by granitic gneisses of the Dalgaringa Supersuite, which are consistent with formation in a supra-subduction zone setting[35–37]. North-dipping subduction was partly simultaneous with the first stage of the Glenburgh Orogeny (2.005–1.985 Ga), and ended with the closure of an ocean basin and collision with the Yilgarn Craton to the south during the second stage of the Glenburgh Orogeny at 1.965–1.950 Ga, forming the West Australian Craton[33,38]. The northern margin of the Narryer Terrane was deformed during this collision, reworked in the subsequent 1.82–1.77 Ga intracratonic Capricorn Orogeny, and is now represented by the Yarlarweelor Gneiss Complex. The Errabiddy Shear Zone, which marks the present-day boundary between the Glenburgh Terrane and the Yilgarn Craton, formed during the Glenburgh Orogeny and was affected by dextral transpression during the Capricorn Orogeny, which also resulted in the intrusion of large plutons into, and north of, the Glenburgh Terrane[39].

Here we present results from a seismic survey across the Archaean Yilgarn Craton and the adjacent Palaeoproterozoic Capricorn Orogen to the north that reveal the presence of south-dipping reflections that extend to at least 20 s, i.e. to depths of at least 60 km. We suggest that the most likely interpretation is that these reflections were created by subduction of an oceanic plate as early as 2.82 Ga in the late Mesoarchaean, up to 130 Ma earlier than implied by previous seismic reflection surveys, but we cannot exclude the possibility that the reflections may have arisen through deep underthrusting, i.e. limited subduction[40], of continental crust during assembly of the West Australian Craton at 1.965–1.950 Ga and/or the subsequent intracratonic Capricorn Orogen at 1.82–1.77 Ga.

## Results

**West Australian seismic reflection survey.** In 2010, Geoscience Australia and the Geological Survey of Western Australia acquired deep seismic reflection profiles (Fig. 1) across the northern Youanmi Terrane, the Capricorn Orogen to the north, and the Southern Carnarvon Basin to the west[41,42]. For our study, line 10GA-YU1 was reprocessed to enhance lower amplitude reflections from the uppermost mantle; in this processing (see Methods section), the 3D orientation of reflectors was estimated, with the dip and strike parameters used to compute an improved stack section[43]. This reprocessed stack was appended to the original stack of line 10GA-CP3, and migrated (Fig. 2a). (We subsequently abbreviate the names of the seismic lines to their last three characters). The migrated composite seismic section reveals the subsurface geometry of a major suture zone with south-dipping reflections extending to depths of at least 60 km where there is a dramatic northward increase from 35 km to ~50 km in the depth of the Moho, which is mostly well defined and inferred from the downward termination of seismic reflections.

In the Youanmi Terrane, the upper crust is relatively transparent where large granites are mapped at the surface, and the middle and lower crust are characterized by shallowly dipping and sub-horizontal reflections. A small number of isolated reflections appear to correlate with outcropping Proterozoic sills, and cut across the pervasive mid-crustal reflectivity, which is

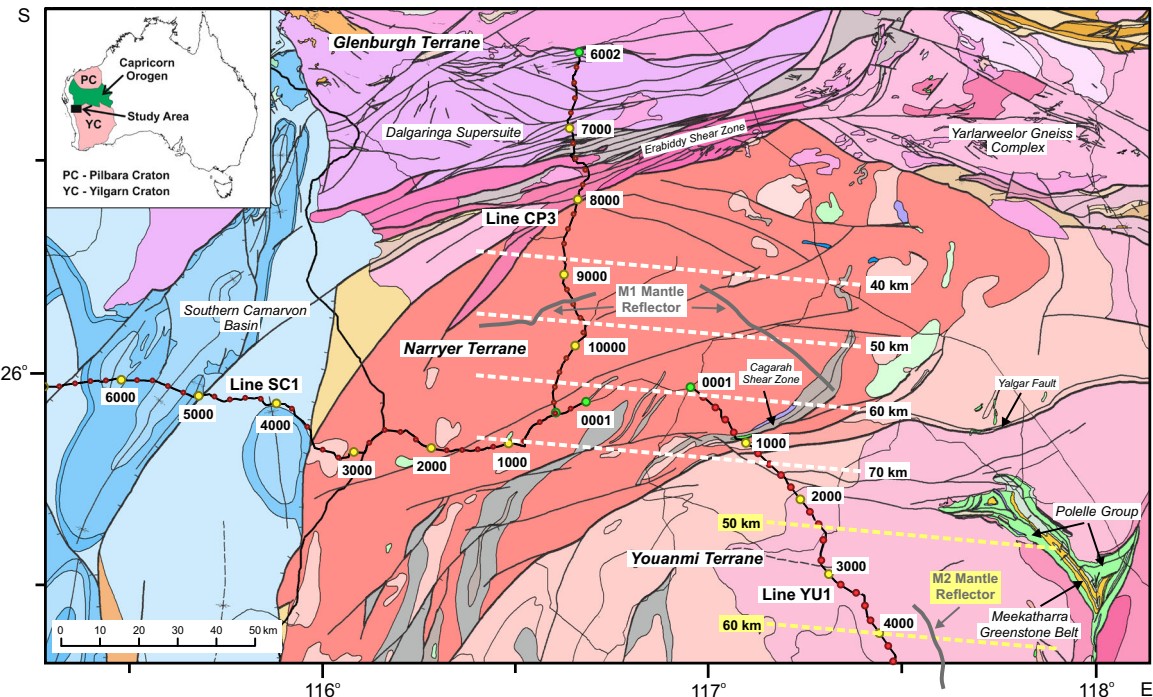

**Fig. 1 Geology of the northwest Yilgarn Craton showing the location of seismic profiles.** The Narryer Terrane comprises exhumed amphibolite and granulite facies gneissic rocks of the northwest margin of the Archaean Yilgarn Craton, the core of which is the Youanmi Terrane. The Glenburgh Terrane accreted to the Yilgarn Terrane during formation of the West Australian Craton at 1.965–1.950 Ga, and the subsequent Capricorn Orogen reworked the collision zone, including the northern part of the Narryer Terrane now represented by the Yarlarweelor Gneiss Complex. Dashed white lines—depth contours of the planar interface that approximates the M1 mantle reflector, dashed yellow lines—depth contours of the planar interface that approximates the M2 mantle reflector, solid grey lines—reflection points on the interfaces, common depth points (CDP) are indicated every 200 by filled circles along the seismic lines. Overview of geological units: blue—Palaeozoic sedimentary rocks, orange/brown – Proterozoic sedimentary rocks, grey—Proterozoic and Archaean metasedimentary rocks, green/yellow—volcanic greenstone rocks, pink/red—granites and granitic gneiss.

inferred to be Archaean in age[44,45]. The lower crustal unit immediately above the Moho, which has been previously referred to as the Yarraquin Seismic Province[46], can be readily followed from line YU1 to line CP3, where it is deflected downward and appears to intersect the higher amplitude package of south-dipping mantle reflections at Common Depth Point (CDP) 9300. North-dipping reflections project to the surface near CDP 800 on line YU1, which is the location of the Jack Hills greenstone belt. These reflections cannot be clearly traced from line YU1 to line CP3, but they do project downward to a zone of disrupted reflectivity that we interpret to be the base of the Narryer Terrane, which is internally characterized by a complex pattern of seismic reflectivity (Fig. 2b). The north end of line CP3 reveals a strikingly different structural style with two 2 s thick packages of south-dipping reflections that extend from the upper and middle crust to near the base of the crust[47], which is less well defined than the Moho below the Youanmi Terrane.

On the unmigrated stack of line YU1, a 1 s thick zone of reflections can be followed from 14.5 s at the north end of line YU1 to the maximum recording time of 20 s (Fig. 3a and Figs. S1 and S2), giving rise to the dipping mantle reflections that appear to intersect the Moho on line CP3 after migration. Local prestack estimates of reflector orientations, which are computed for each CDP from 64 adjacent common depth point gathers, indicate a range of strike azimuths for the mantle reflector of 090° to 120° (Fig. 3c); however, these estimates are only reliable in a few locations due to the low range of source-receiver azimuths in the straight sections of the seismic profile. Complementary zero-offset forward modelling of the reflections under the assumption that they originate from a planar dipping interface for the crooked geometry of line YU1 indicates that the mantle reflector

has a fairly limited range of possible orientations: dip of $29 \pm 2°$ at a strike of 090° to $42 \pm 2°$ at a strike of 105° (Fig. 3a); a strike azimuth of 095° is most consistent with the limited mantle reflectivity observed on east-striking line SC1 (Fig. S3). Another band of dipping reflections (Figs. S1 and S2), which after migration corresponds to M2 in Fig. 2a, originates in the mantle beneath the Youanmi Terrane, and has a dip of 18–21° at a strike of 75–105°, though this orientation is less well determined due to its shorter lateral extent. In summary, we identify two bands of approximately south-dipping reflectors below the northwest Yilgarn Craton (Fig. 1), which likely extend to greater depths, because the reflections are observed to the maximum recording time of 20 s.

**Interpretation.** The major terrane boundaries crossed by the seismic profile have experienced a prolonged deformational history, and were strongly affected by dextral transpression during the Capricorn orogeny[47,48]. We interpret the Narryer-Youanmi terrane boundary at depth to correspond to S1 (Fig. 2b), a ~5 km wide zone across which the dip of mid-crustal reflections reverses. Curved reflectors adjacent to S1 may be the remnants of folds created during the collision of the Narryer and Youanmi terranes, but their disruption within S1 is likely due to synchronous or subsequent transpression. In the lower crust, the surface along which the Narryer Terrane was thrust over the Youanmi Terrane is inferred to be T1 at the top of the reflective lower crust, i.e. the Yarraquin Seismic Province. The subsurface extent of the Narryer Terrane is constrained by following downward reflections that project to its surface outcrop; where this is not possible our interpretation is shown as less certain, though we note that reworked Narryer rocks of the Yarlarweelor complex project

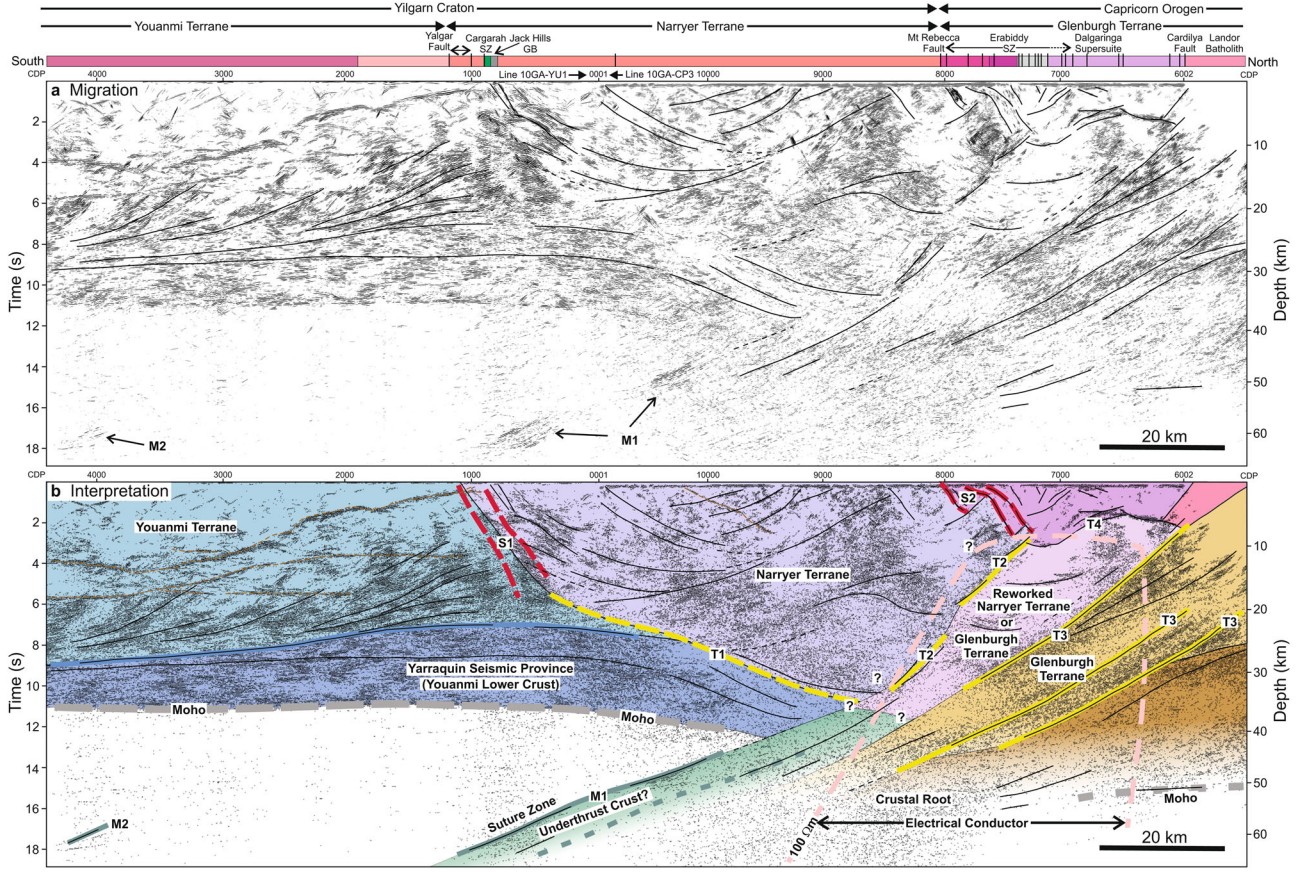

**Fig. 2 Seismic reflection section showing lines YU1 and CP3. a** Segment migration[64] of combined seismic section constructed from seismic lines YU1 and CP3. Prominent reflections, and the boundaries between major reflective units are interpreted. Overview of geological units: grey—Proterozoic and Archaean metasedimentary rocks, green—volcanic greenstone rocks, pink/red—granites and granitic gneiss. **b** Kirchhoff migration superimposed on interpretation of major crustal blocks and the shear zones related to their evolution. Dashed pink line—100 Ω.m iso-resisitivity contour around a steeply dipping to subvertical conductor associated with the boundary between the Glenburgh and Narryer Terranes. Dashed brown lines—interpreted sills, dashed thick green line—possible more northerly position of M1 along line CP3.

along strike as far north as CDP 6000 on line CP3 (Fig. 1), consistent with the terrane interpretation shown in Fig. 2b.

The Erabiddy Shear Zone, which is mapped as a corridor of anastomosing north-dipping shear zones, separates the Narryer Terrane from the Glenburgh Terrane[47] in the upper crust, and we denote the southern edge of this shear zone as the northern limit of the Narryer Terrane at depth (S2 in Fig. 2b). Where identifiable most near-surface reflectors in this corridor are north dipping, and it appears to be a complex imbricated zone into which the southernmost rocks of the Dalgaringa Supersuite are also incorporated[47]. To the north of the Erabiddy Shear Zone, rocks of the Dalgaringa Supersuite are underlain at 5 km by a strong reflector (T4 in Fig. 2b) that deepens to the south, and this may indicate a thrust along which Dalgaringa rocks were transported during Proterozoic transpression. We interpret the prominent south-dipping packages of reflectors that underlie the Dalgaringa Supersuite as panels of the Glenburgh Terrane that were thrust beneath the Narryer Terrane during the Glenburgh and Capricorn orogenies[47]. We speculate that T2 represents an earlier thrust fault that was subsequently disrupted by transpression during the Capricorn Orogeny, which imbricated much of the crust along moderately dipping shear zones, e.g., T3.

At depths >40 km between CDP 7500 and CDP 10000 on line CP3, there is a zone of diffuse reflectivity that mostly exhibits apparent dips to the south (Fig. 2a), and this region is likely to be underthrust crust of the Glenburgh Terrane[47] in which eclogitization may have reduced the amplitude of the

deeper seismic reflections[49]. The shallower more prominent band of reflections M1 at 14–18 s, which were recorded on line YU1, have migrated to the position shown in Fig. 2 under a 2D assumption; however, since these reflections originated out-of-plane, their relation to the northward termination of the shallower section of Moho and the crustal root is not completely clear. It is possible that these reflections form the upper part of the underthrust root or they could be located further south, away from the suture zone.

## Discussion

The Yilgarn Craton possesses a lithospheric mantle root characterized by relatively high seismic velocities that extend up to 250 km depth[50,51]. At depths >60 km in the vicinity of the seismic reflection profiles, the boundary between this high-velocity root and lower velocity mantle to the north appears to be subvertical[52], which is consistent with a steep south-dipping conductor imaged to at least 60 km depth below the southern limit of the Glenburgh Terrane (Fig. 2b) by a long-period magnetotelluric survey[53]. If this is an accurate representation of the edge of the high-velocity root at 40–60 km depth, then seismic reflectors M1 and M2 lie within the Archaean lithospheric mantle. Alternatively, if the northern edge of the mantle root dips more shallowly to the south, reflector M1 probably represents the boundary between the Archaean lithospheric mantle associated with the Youanmi Terrane and Proterozoic lithosphere. We consider two alternative interpretations consistent with the seismic data: (a) the reflections

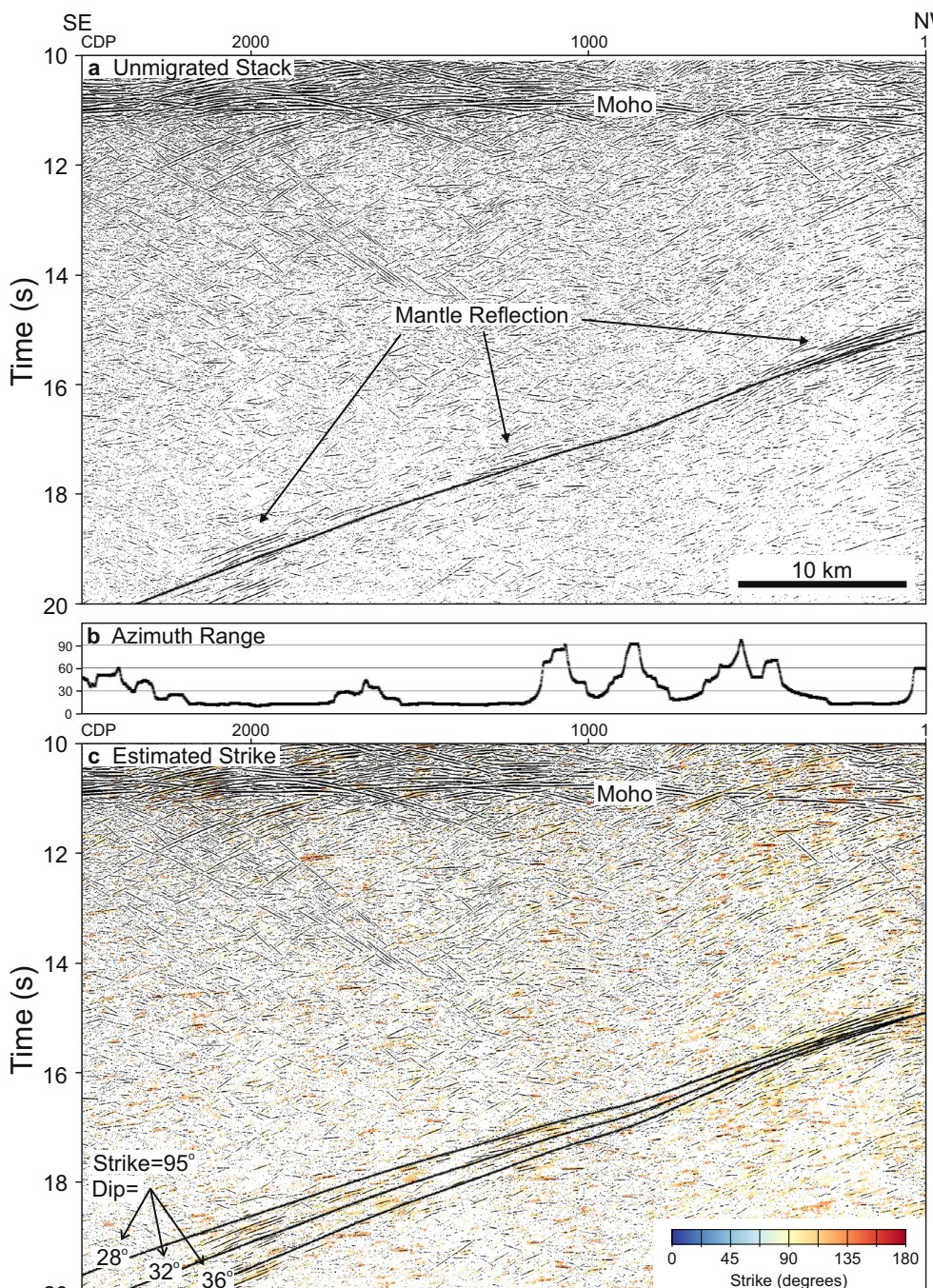

**Fig. 3 Mantle reflections M1. a** Unmigrated stack of line YU1. Superimposed solid lines, which represent the nearly identical travel times of three planar reflectors with dip/strike of 29°/090°, 32°/095°, and 36°/100°, all vary in a similar fashion to the mantle reflections. Smaller and larger values of strike cannot be fit as well to the observed reflections. **b** Range of source-receiver azimuths included in the estimation of dip and strike along line YU1. This range is small where the line is almost straight. **c** Local prestack estimates of reflector strike where they lie between 090° and 120° superimposed on the stack section indicating that most strong mantle reflections exhibit strike values in this range. Superimposed solid lines indicate the travel times of three planar reflectors with a strike of 095°, and dips of 28°, 32°, and 36°, from which we infer that for a given strike the dip of the best fitting planar interface can be determined to within 2°.

arise from a suture zone within Archaean lithosphere created by the subduction of oceanic crust beneath the northern Youanmi Terrane at >2.82–2.74 Ga, which ultimately resulted in the collision with the Narryer Terrane; (b) the reflections arose during deep thrusting of the continental Glenburgh Terrane beneath the Yilgarn Craton between 1.965 Ga and 1.77 Ga during the Glenburgh and Capricorn orogenies. The absence of Palaeoproterozoic arc-related rocks on the northwest Yilgarn Craton and the

well-defined lower crust and Moho, which were established under the Youanmi Terrane late in the Neoarchaean[54] and have not been subsequently disrupted by arc magmatism, preclude a south-dipping subduction zone along the edge of the Yilgarn Craton in the Palaeoproterozoic.

The greenstone sequences in the northwestern Yilgarn Craton contain boninite-like rocks within the lower parts of both the 2.82–2.80 Ga Norie Group and the 2.80–2.74 Polelle Group

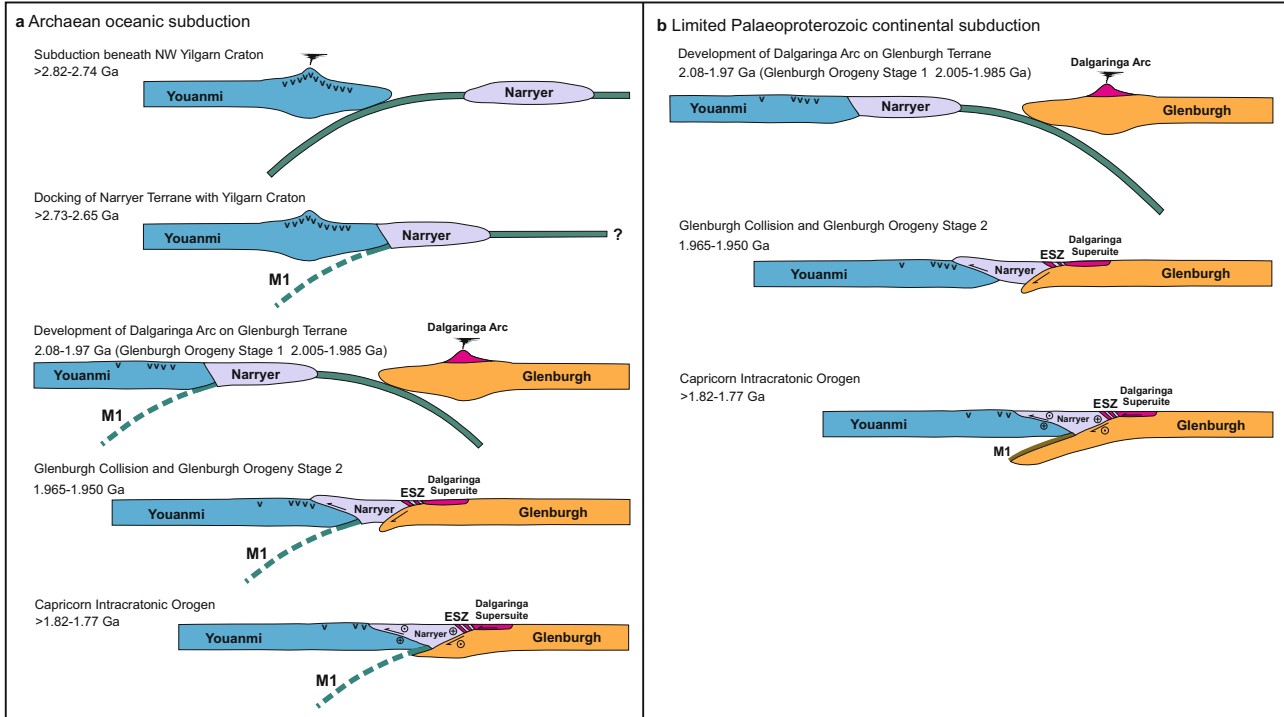

**Fig. 4 Summary of two alternative interpretations of the evolution of the northwest margin of the Yilgarn Craton. a** Subduction beneath the northwest Youanmi Terrane of the Yilgarn Craton, with the related emplacement of boninitic rocks and sanukitoids, terminated with docking of the Narryer Terrane, and left a remnant scar in the uppermost mantle. The two-stage collision of the Glenburgh Terrane with the Yilgarn Craton[36,47] created the Erabiddy Shear Zone (ESZ), and further obducted the Narryer Terrane onto the lower crust of the Youanmi Terrane, a process which largely concluded with the intracratonic Capricorn Orogen. **b** During assembly of the West Australian Craton the Glenburgh Terrane was thrust beneath the Narryer Terrane, which had previously docked with the Youanmi Terrane during the Neoarchaean. In the subsequent Capricorn Orogen, the Glenburgh Terrane was partly subducted beneath the edge of the lithospheric root of the Yilgarn Craton, creating the seismic reflectors that mark the top of deeply underthrust continental crust. The obduction of the Narryer Terrane largely terminated at this time. v—indicates Archaean volcanic rocks with arc affinity.

(Fig. 1), which are associated with high-Mg andesites, sanukitoids and hydrous mafic intrusions; the presence of these boninitic rocks is best explained by volcanism in the forearc of some form of, perhaps short-lived or intermittent, subduction zone[25,26], which may also have driven accretion of the Narryer Terrane[22,27]. If reflector M1 lies within mantle of Archaean age, then we suggest that these approximately south-dipping reflectors were created to depths of at least 60 km by Archaean subduction of an oceanic plate, and that the boninitic rocks were erupted on the northwest margin of the Youanmi Terrane, which represented the forearc of this subduction zone[22] (Fig. 4). With a moderate dip of ~32° and a planar geometry imaged over a lateral distance of 40 km, i.e., showing no evidence for buckling or imbrication, reflector M1 exhibits none of the features that might be expected to arise from shallow underthrusting of buoyant, relatively weak oceanic crust in a presubduction setting. It is possible that reflector M2 is the remnant of an earlier phase of subduction, which is consistent with the observation of two cycles of volcanism that produced boninite-like rocks[25], and these two episodes of subduction may have led to the growth of the mantle lithosphere by slab imbrication following step back of the subduction zone[55]. Though delamination of a thickened crust by peeling back of its dense eclogitic base has the potential to create dipping reflections in the uppermost mantle, such a process could not produce the hydrous volcanic rocks found at the surface. The lack of observation of similar dipping reflectors in the interior of the Youanmi Terrane indicates that subduction may only have influenced the margin of this terrane. Since no subduction-related rocks have been identified on the Narryer Terrane, subduction, which was probably a relatively unstable process at that time[7],

must have terminated or perhaps reversed polarity rather than stepping back when the Narryer Terrane collided with the Youanmi Terrane (Fig. 4). The intrusion of granites into both the Narryer and Youanmi terranes at 2.66 Ga indicates that these two terranes were in contact by this time; however, 1.82–1.77 Ga dextral transpression and deformation along the Cagarah shear zone and the Yalgar fault (Fig. 2b), which extends into the Youanmi Terrane, indicate that obduction of the Narryer Terrane onto the lower crust of the Youanmi Terrane was not complete until following the Capricorn Orogen[28].

If dipping reflections M1 are not related to Meso- to Neoarchaean assembly of the Yilgarn Craton, then they must have arisen during the Palaeoproterozoic Glenburgh and/or subsequent Capricorn orogenies. Other examples of dipping seismic reflections in the upper mantle have been tied to subduction of an oceanic plate prior to terrane collision, but the well-defined lower crust and absence of Palaeoproterozoic arc-related rocks in the northwest Yilgarn Craton indicate that there was no south-dipping subduction zone along the edge of the Yilgarn Craton at this time. The location on the Glenburgh Terrane of arc-related rocks of the Dalgaringa Supersuite and older detrital zircons indicate that a north-dipping subduction zone existed between the Yilgarn Craton and Glenburgh Terrane from 2.08 Ga to 1.97 Ga[38]. With the closure of the ocean basin, the Glenburgh Terrane was thrust below the Narryer Terrane, though the magnitude of the crustal shortening at this time is unclear; however, in this scenario, crustal rocks of the Glenburgh Terrane are thrust to depths of at least 60 km beneath the northern edge of the strong, relatively buoyant Archaean lithosphere of the Yilgarn Craton by the end of the Capricorn Orogeny. In this case,

subparallel reflector M2 could be a shear zone created further south within Archaean lithospheric mantle by Proterozoic shortening.

Though the similarity in the strike of the mantle reflectors and mapped faults related to Proterozoic transpression might seem to favour the deep underthrusting of the Glenburgh Terrane, the orientation of later structures may be inherited, because the edge of an Archaean craton can act as a zone of shear localisation controlling the geometry of subsequently accreted terranes and the development, and even polarity reversal, of subduction zones during the Palaeoproterozoic, as appears to have occurred along the western edge of Slave Craton[14]. Given the steep dip currently inferred for the northern edge of the high-velocity lithospheric mantle near the seismic reflection profiles and the occurrence in the northwest Youanmi Terrane of Archaean rocks with an arc-affinity, we favour an interpretation of Archaean oceanic sub-duction. The lack of Proterozoic rocks with an arc-affinity within the Narryer Terrane, together with the lack of evidence for modification of the underlying lower crust through even incipient arc magmatism seem to preclude Palaeoproterozoic subduction as an alternative model unless extremely short-lived.

Fundamental changes in the geological record, including the appearance of passive margins[56], arc volcanic rocks such as boninites[22,57], and paired metamorphic belts[1,58] have been used to suggest that plate tectonics and subduction began during the Mesoarchaean[59–61], and had become the dominant tectonic regime by the end of the Archaean[62]; however, the timing of the onset of plate tectonics remains controversial[63]. Our favoured seismic interpretation pushes the evidence for subduction infer-red from seismic reflectors in the upper mantle, which are also linked to horizontal shortening in the overlying crust, back to >2.8 Ga, representing the first example from the Mesoarchaean. We recognise, however, that further observations to define better the downward continuation of the Yilgarn mantle reflections and the edge of the high-velocity Archaean lithospheric mantle are required to characterise the extent of this process and to distinguish conclusively between our alternative interpretations.

## Methods

**Line YU1**. Line YU1 was reprocessed to enhance deep reflections, using refraction static and residual static corrections calculated for the original processing, which had been carried out at Geoscience Australia[42]. CDP bins were assigned to a smoothed slalom line through the acquisition profile, and evenly incremented every 20 m along an azimuth of 145° from north. Prestack processing included geometry assignment, static corrections, amplitude recovery, spectral whitening over a bandwidth of 6–60 Hz, automatic gain control with a 0.5 s window, and muting. At every CDP location and time, the dip and strike values of the most coherent reflection were determined[43] from a supergather of the 64 adjacent CDP. The stack was computed using 3D normal moveout, which accounts for a reflector's dip and strike, and a correction of the midpoint to the bin centre based on these orientation values. A comparison of this unmigrated stack section with a stack obtained with the same processing flow, but a 2D normal moveout correction, is shown in Fig. S1. The benefit of the 3D reflector orientation information and incorporating data from adjacent CDPs is apparent both before and after post-stack coherency fil-tering. It should be noted that use of a dip moveout (DMO) correction in the conventional processing flow would improve the visibility of shallow reflections, but would have no significant effect at late times where the mantle reflections are observed. Use of cross-dip corrections has the potential to improve reflection stacking, but is difficult to apply where reflections have different 3D orientations. Though the highest amplitude part of mantle reflection M1 is visible at the northwest end of line YU1 after coherency filtering of the original processing, this reflection does not appear in the migration, because it moves beyond the end of the seismic profile.

**Line CP3**. To match the lateral smoothing implicit in the reflector orientation processing of line YU1, the stack of line CP3 was subject to coherency enhance-ment using summation over 21 traces along the most coherent dip at each time sample, e.g., Fig. S4, and then appended to line YU1; lower crustal reflections correlate well between the ends of the two lines. Trace amplitudes in the combined stack were equalised using a time window of 14–19 s, which is later than the high amplitude crustal reflections that might bias estimated amplitude values. The

composite section was then migrated using an algorithm that repositions reflec-tions in the data based on their apparent dip, in a similar fashion to the line migration of interpreted reflectors, which does not generate wave-equation artefacts[64].

**Line SC1**. Line SC1 was reprocessed in the same fashion as line YU1, but the refraction statics were computed from a velocity model derived by tomographic inversion of picked first arrivals.

## Data availability

Seismic reflection data for lines YU1, CP3, and SC1 are available in SEGY format from Geoscience Australia (https://www.ga.gov.au/about/projects/resources/seismic/wa-datasets). The geological map was constructed using information available through GeoView.WA (https://geoview.dmp.wa.gov.au/geoview/?Viewer=GeoView), the interactive geological map of the Geological Survey of Western Australia.

## Code availability

Seismic reflection processing was carried out using TomoPlus and ProMAX software available from GeoTomo and Halliburton Corp respectively under commerical licensing arrangements; additional code to compute reflector orientations and segment migration are available from the authors as subroutines for ProMAX version 5000.0.3 on reasonable request.

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

## Acknowledgements

Ross Costelloe and Leonie Jones carried out prestack seismic processing of lines YU1 and CP3. We thank Dave Champion, Dennis Brown, and Don White for helpful comments. The field acquisition was funded by Geoscience Australia, the Geological Survey of Western Australia, and AuScope. This project was supported by the Natural Sciences and Engineering Council of Canada through grant RGPIN-04185 to AC. MD publishes with the permission of the CEO, Geoscience Australia. This contribution forms part of Geoscience Australia's Exploring for the Future Program.

## Author contributions

A.C. and M.D. interpreted the seismic data and wrote the paper. S.S. carried out processing for line SC1.

## Competing interests

The authors declare no competing interests.
