## [Peer Review File · Nature Communications]

Reviewers' Comments:

Reviewer #1:

Remarks to the Author:

Calvert, Doublier and Sellars report a significant new finding in their submission "Seismic reflections from a lithospheric suture zone below the Archaean Yilgarn Craton". There are only a few examples of such deep dipping reflectors associated with cratons world-wide, and the great majority do not extend ~ 60 km into the mantle. The authors have done a good job of presenting a balanced paper that provides two plausible explanations for this feature. Either scenario is very noteworthy, and I do not consider the fact that the results may be explained by alternative models to be a weakness of the paper in terms of its worthiness for publication in Nature Communications. I spent most of my review time thinking about whether the authors had missed some critical piece of information that would undisputedly distinguish between the alternate scenarios, but I could not come up with anything.

The work has significance for our understanding of cratons globally and for the Yilgarn Craton in particular. While the authors raise the possibility of a second slab origin for the M2 reflector, they cite only "generic" publications concerning short-lived Archean subduction. I suggest that they specifically cite Lowrey et al (ref 17) in this regard because that paper accounts for two cycles of Youanmi (north-western Yilgarn) volcanism (each with early boninites or boninites-like rocks) by a process such as subduction step back. Beyond that, I cannot find anything in the paper that is in need of revision. The methodology appears first rate.

The authors state at lines 172-176 that there is no evidence for a south-dipping slab beneath the Yilgarn in the Paleoproterozoic and I would agree. They also cite Cook et al. (ref 5) regarding the potential edge effects that cratons can impart on accreted terranes. In a paper of this type, space constraints likely prevent further detailed discussion along these lines. I note, however, the potential significance of craton edge effects and the broader implications of this paper. The 2.0 Ga upper amphibolite Glenburgh gold deposits in the Dalagrainga Supersuite (Fig 4) are considered "orogenic" (p. 82 of Roche et al., 2017, Precamb. Res. 290 63-85) and appear to have derived some of their Sulphur, etc, from the Narryer (Selvaraja et al., 2017, Geology 45, 419-422). Based on most orogenic Au models, then, the Glenburgh could not have been underthrust beneath the Narryer in the first stages of the orogeny (peak metamorphism at c. 1991 Ma: Roche et al.). Cawood and Tyler (2004, Precamb Res 128, 201-218), however, distinguish two stages to the Glenburgh orogeny (2000 – 1970 Ma and < 1975 Ma – > 1965 Ma). In order for the orogen to somewhat resemble that shown in Fig 4, it seems likely that the Glenburgh "wedged" the Narryer, during stage 2 of the orogeny. It also seems possible that the south-dipping Narryer-Yilgarn Archean subducted slab has been "enhanced" by the Yilgarn wedging into the Narryer and peeling back the lower crust. In any case, these are the types of issues that the new observations of Calvert and co-workers bring into focus and the paper will clearly stimulate much discussion along these lines.

In summary, this paper presents exciting new results that appear (to me) to confirm specific geodynamic predictions made based on studies of NW Yilgarn volcanism. While some ambiguity remains, the results presented here will contribute to our understanding of Precambrian sutures globally and will promote new avenues of investigation in the West Australian Craton. I fully support the acceptance of this paper for publication in Nature Communications.

Reviewer #2:

Remarks to the Author:

Upper mantle reflections (lithospheric) have been the subject of many crustal-scale seismic studies including those from Lithoprobe-Canada and BABEL-Fennoscandia. This manuscript is not an exemption and covers

similar topics. While upper mantle reflections in the Archean terrane may seem rare, there are not so many Archean terranes around the world to allow such a deep crustal study nor there are many such deep seismic studies across them. Hence, the observation of lithospheric reflections may not necessarily be unique in the sense that they are not observed, it is just a matter of time until more studies and proper seismic profiles are available. Reprocessing of historical data is one way of revealing these reflections as was also recently documented from reprocessing offshore BABEL lines by Buntin et al. (2020) although from a Paleoproterozoic setting (surface geology).

I have missed to figure out what were the interpretation of the original processing works and how that differ with the current interpretation. Most of the south-dipping package of reflectivity are within CP3 profile and only YU1 was reprocessed. So, I am assuming these sets of reflections were there in the original processing work. It would be good to see this is clarified also comparisons between the original and reprocessed works are shown in the supplementary information and details of how the reprocessing workflow led to an improved section.

I am not convinced that such a deep set of reflectivity and of such a nature (2 s wide) would require any strike analysis or even sensitive to any cross-dips. They will likely be imaged with any strike and dip given the wavelength involved at such a depth.

I am also confused by the wording in the manuscript of how the two sections were merged. My understanding is that only YU1 was reprocessed up to unmigrated stack and then stitched to CP3 and then migrated together. How were the amplitudes balanced?

It is unclear what "migration free artefact" algorithm is in the text? Did you mean line-drawing migration or could you clarify? Perhaps this also connects to the comment above.

Apart from these comments, my main concern is to not seeing any clear information on the original interpretations and processing works of the datasets and not sure how significant and novel is this work since the upper mantle reflectors from the same leading author was already reported in the earlier studies from the Lithoprobe seismic profiles in Archean settings but also from the Paleoproterozoic settings.

Reviewer #3:

Remarks to the Author:

The manuscript is an interesting contribution in deep seismic lithosphere structures and plate tectonic processes responsible of them. The authors study the Yilgarn craton with three seismic 2D lines reprocessed for the study of upper mantle reflectors. The authors report a south-dipping reflector extending up to about 60 km depth in the mantle. Extrapolated to surface the structure sits between approximately between the <2.555 Ga Glenburgh and the 3.7-3.0 Ga Narryer terranes.

The authors provide two alternative interpretations for the reflector. According to the first model, it would represent a frozen-in remnant of Archaean subduction zone (preferred by the authors), and according to the second model, it would be a suture from thrusting of the Glenburgh terrane beneath the Yilgarn craton in the Proterozoic.

I have the following notes and comments on the manuscript.

(1)The manuscript lacks a good introductory paragraph pointing out what are the implications of the study beyond the Yilgarn craton evolution. Are we just reading a report on seismic reflectors in Yilgarn, or do we see a thorough discussion on the first plate tectonic subduction process? This should be improved.

(2) The authors seem to assume that Archaean and Paleoproterozoic plate tectonic processes were similar to the present ones. Thus, they simply follow the uniformitarian principle, which is not actually correct when interpreting Archaean processes. The geothermal conditions were quite different in Archaean and Palaeoproterozoic times due to much higher radiogenic heat production, higher mantle heat flow, and higher geothermal gradient. The tectonic mechanism must have operated differently at least until end of early Archaean. The plates were much thinner, and it is still an open question when did plate tectonics start to operate in the modern sense, although ~3 Ga is often mentioned. Due to the hot lithosphere in the continental and oceanic areas, subduction may have not been possible at all until the earth had cooled sufficiently. As we know, subduction requires a positive density contrast between the slab and the upper mantle, and such a contrast may not have existed if the oceanic crust is assumed to have been basaltic. On the other hand, subduction may have started to operate early if the oceanic crust comprised high Mg-basalts (i.e., komatiites) due to high potential temperature of the mantle. Therefore, the authors have here an opportunity to indirectly imply something about the geothermal conditions and plate tectonic processes during the Archaean.

(3) The interpretation of the Yligarn mantle reflector as a suture zone is naturally the most plausible interpretation, especially when the geological evidence is considered, and similar seismic structures have been presented in literature for both Archaean and Proterozoic areas with similar interpretations. However, taking into account the hot geothermal conditions, we should also consider different mechanisms, which would have produced similar structures. The geological character of the mantle reflector remains speculative, of course, but most probably it is due to eclogitized basaltic rock or shearing. If the terranes in Yilgarn had eclogitic lower crust already in the Archaean (?) it would have allowed removal of the lowermost eclogitic crust by delamination in the orogenic peak temperatures, either as a slab or as a fluid, i.e. Rayleigh-Taylor-style flow over geologically short times scales (~10-30 Ma). Lower crustal mafic rock would have turned into eclogite at early stages of crustal thickening in collisions of arc-type crusts and plates. In the present conditions Moho is at >50 km under the Glenburgh terrane, and taking into account the layers removed by erosion (10-15 km?) it may have been much deeper initially.

(4) There are stack sections combining and comparing survey lines CP3 and YU1 (Fig. 2) and SC1 and YU1 (Suppl. Fig. 2). I would be keen to see also a figure showing SC1 and CP3. This would help in 3D visualization of the reflector.

Aug. 20, 2021
Ilmo Kukkonen

Response to Reviewers

We sincerely appreciate the time and effort that the reviewers and Senior Editor have invested in handling our paper, especially in these challenging times. We list in **red** our response to the reviewers' comments immediately following the paragraphs to which they apply.

Reviewer #1 (Remarks to the Author):

Calvert, Doublier and Sellars report a significant new finding in their submission "Seismic reflections from a lithospheric suture zone below the Archaean Yilgarn Craton". There are only a few examples of such deep dipping reflectors associated with cratons world-wide, and the great majority do not extend ~ 60 km into the mantle. The authors have done a good job of presenting a balanced paper that provides two plausible explanations for this feature. Either scenario is very noteworthy, and I do not consider the fact that the results may be explained by alternative models to be a weakness of the paper in terms of its worthiness for publication in Nature Communications. I spent most of my review time thinking about whether the authors had missed some critical piece of information that would undisputedly distinguish between the alternate scenarios, but I could not come up with anything.

The work has significance for our understanding of cratons globally and for the Yilgarn Craton in particular. While the authors raise the possibility of a second slab origin for the M2 reflector, they cite only "generic" publications concerning short-lived Archean subduction. I suggest that they specifically cite Lowrey et al (ref 17) in this regard because that paper accounts for two cycles of Youanmi (north-western Yilgarn) volcanism (each with early boninites or boninites-like rocks) by a process such as subduction step back. Beyond that, I cannot find anything in the paper that is in need of revision. The methodology appears first rate.

Thanks for making this point, which is a valuable additional observation, and we now explicitly note the possible connection between the two mantle reflectors and the two distinct cycles of volcanism in the Youanmi Terrane, citing Lowry et al. (2019).

The authors state at lines 172-176 that there is no evidence for a south-dipping slab beneath the Yilgarn in the Paleoproterozoic and I would agree. They also cite Cook et al. (ref 5) regarding the potential edge effects that cratons can impart on accreted terranes. In a paper of this type, space constraints likely prevent further detailed discussion along these lines. I note, however, the potential significance of craton edge effects and the broader implications of this paper. The 2.0 Ga upper amphibolite Glenburgh gold deposits in the Dalagrainga Supersuite (Fig 4) are considered "orogenic" (p. 82 of Roche et al., 2017, Precamb. Res. 290 63-85) and appear to have derived some of their Sulphur, etc, from the Narryer (Selvaraja et al., 2017, Geology 45, 419-422). Based on most orogenic Au models, then, the Glenburgh could not have been underthrust beneath the Narryer in the first stages of the orogeny (peak metamorphism at c. 1991 Ma: Roche et al.). Cawood and Tyler (2004, Precamb Res 128, 201-218), however, distinguish two stages to the Glenburgh orogeny (2000 – 1970 Ma and < 1975 Ma – > 1965 Ma).

We have changed the text to explicitly refer to the two stages of the Glenburgh Orogeny, and modified Fig. 4 to refer to the two stages accordingly. We have included the dates for Stage 1 of the Glenburgh

Orogeny in Fig. 4 with north-dipping subduction at the time of mineralisation and arc magmatism, and show Stage 2 when the Glenburgh Terrane was thrust beneath the Narryer Terrane in the panel below . The dates we quote are from the work summarised in Johnson et al. (2011a,2011b), which are based on the earlier work by Cawood and Tyler (2004). In the model of Selvaraja et al. (2017), sediments shed from the Narryer Terrane are subducted beneath the Glenburgh Terrane, and we now cite this paper as further evidence of a north-dipping subduction zone between the Narryer and Glenburgh terranes.

In order for the orogen to somewhat resemble that shown in Fig 4, it seems likely that the Glenburgh “wedged” the Narryer, during stage 2 of the orogeny. It also seems possible that the south-dipping Narryer-Yilgarn Archean subducted slab has been “enhanced” by the Yilgarn wedging into the Narryer and peeling back the lower crust. In any case, these are the types of issues that the new observations of Calvert and co-workers bring into focus and the paper will clearly stimulate much discussion along these lines.

We don't identify any specific wedging of the Glenburgh Terrane into Narryer Terrane, or major disruption of the Narryer Terrane, and consider the Glenburgh to be primarily thrust below the Narryer, as did Johnson et al. (2013) in their seismic interpretation; however, rocks of the Narryer and Dalgaringa Supersuite are shown imbricated during Stage 2, possibly during transpression. In our model, the Narryer Terrane docks with the Youanmi Terrane before being mostly obducted onto the Yilgarn Craton. It is possible that wedging of the Youanmi lower crust into the Narryer Terrane separated part of the lower Narryer crust, forcing it down into the mantle. We are, however, unable to distinguish between the presence of some lower continental crust or oceanic crust at these depths, and have just referred to this region as “crust” without attributing a specific origin. We hope future work on ground truthing aspects of our interpretation will help clarify the nature of some of the contacts and processes we have inferred.

In summary, this paper presents exciting new results that appear (to me) to confirm specific geodynamic predictions made based on close to the mantle reflector M1 studies of NW Yilgarn volcanism. While some ambiguity remains, the results presented here will contribute to our understanding of Precambrian sutures globally and will promote new avenues of investigation in the West Australian Craton. I fully support the acceptance of this paper for publication in Nature Communications.

Reviewer #2 (Remarks to the Author):

Upper mantle reflections (lithospheric) have been the subject of many crustal-scale seismic studies including those from Lithoprobe-Canada and BABEL-Fennoscandia. This manuscript is not an exemption and covers similar topics. While upper mantle reflections in the Archean terrane may seem rare, there are not so many Archean terranes around the world to allow such a deep crustal study nor there are many such deep seismic studies across them. Hence, the observation of lithospheric reflections may not necessarily be unique in the sense that they are not observed, it is just a matter of time until more studies and proper seismic profiles are available. Reprocessing of historical data is one way of revealing

these reflections as was also recently documented from reprocessing offshore BABEL lines by Buntin et al. (2020) although from a Paleoproterozoic setting (surface geology).

It is certainly true that many Archean areas are yet to be surveyed by seismic reflection profiles, and reprocessing can extract additional information from existing datasets, as has been done with the BABEL seismic profiles, to which we now include citations. It is also worth mentioning that at least in the Yilgarn Craton, which is quite well surveyed by modern seismic reflection profiles, such mantle reflections are not observed elsewhere, including across terrane boundaries that have been interpreted by some as locations of past subduction zones.

I have missed to figure out what were the interpretation of the original processing works and how that differ with the current interpretation. Most of the south-dipping package of reflectivity are within CP3 profile and only YU1 was reprocessed. So, I am assuming these sets of reflections were there in the original processing work. It would be good to see this is clarified also comparisons between the original and reprocessed works are shown in the supplementary information and details of how the reprocessing workflow led to an improved section.

The original processing of line CP3 was interpreted by Johnson et al. (2013), and we cite this paper where our interpretation is similar, which is mostly in the crust of the Glenburgh Terrane. We interpret the deeper structure of the Narryer Terrane differently, and are able to better define its base. However, the main difference in our interpretations arises from our merging of lines YU1 and CP3, which allows the mantle reflections to be identified post-migration. In the original processing, the highest amplitude mantle reflections near the northwest end of line YU1 could not be easily seen on the unmigrated stack section, and these reflections also moved beyond the end of line after migration, essentially becoming lost. We now briefly mention this point in the revised Methods section, and have included a new Fig. S1 to illustrate the differences in the stack sections after different processing flows.

I am not convinced that such a deep set of reflectivity and of such a nature (2 s wide) would require any strike analysis or even sensitive to any cross-dips. They will likely be imaged with any strike and dip given the wavelength involved at such a depth.

Though it is true that the influence of dip and strike on the stacking velocity will be minimal at these late times, the cross-dip correction, which is explicitly incorporated into our method, could have some effect. As mentioned above, we have provided a new Fig. S1 to illustrate the differences between our method and a conventional stack, and discuss the processing in more detail in the Methods section. Some of the improvement in our reprocessed stack also arises from the inclusion of information from 64 adjacent CDP for each output CDP.

I am also confused by the wording in the manuscript of how the two sections were merged. My understanding is that only YU1 was reprocessed up to unmigrated stack and then stitched to CP3 and then migrated together. How were the amplitudes balanced?

Yes, only YU1 was reprocessed, and the two sections were stitched together, which was possible because the shallowly dipping lower crustal reflections matched well, as we now note in the Methods section. We also now explain that amplitudes were balanced using the background noise level at late

times, 14-19 s. To achieve a similar degree of lateral continuity in the stack of line CP3 as obtained on the reprocessed line YU1, a coherency filter was applied to CP3, and the result of this filtering on part of line CP3 can be seen in the new Fig. S4.

It is unclear what “migration free artefact” algorithm is in the text? Did you mean line-drawing migration or could you clarify? Perhaps this also connects to the comment above.

Basically the migration algorithm, which we refer to as segment migration, is a line migration approach that can be used with stacked data, rather than a line-based interpretation of the stack; our algorithm can be viewed as similar to a Kirchhoff migration with the aperture limited to near-zero about the pre-determined apparent dip of each reflector. We have expanded our description in the Methods section, but the method is fully explained in the cited paper.

Apart from these comments, my main concern is to not seeing any clear information on the original interpretations and processing works of the datasets and not sure how significant and novel is this work since the upper mantle reflectors from the same leading author was already reported in the earlier studies from the Lithoprobe seismic profiles in Archean settings but also from the Paleoproterozoic settings.

In response to this comment and also reviewer 3 below, we have added new first and final paragraphs to place our work more clearly in the broader context of constraining the initiation of subduction during the Archean. Our paper pushes seismic reflection evidence for subduction back in time from the Neoproterozoic to the end of the Mesoproterozoic, up to 130 Ma earlier than previously published images from the North American Superior Craton.

Reviewer #3 (Remarks to the Author):

The manuscript is an interesting contribution in deep seismic lithosphere structures and plate tectonic processes responsible of them. The authors study the Yilgarn craton with three seismic 2D lines reprocessed for the study of upper mantle reflectors. The authors report a south-dipping reflector extending up to about 60 km depth in the mantle. Extrapolated to surface the structure sits between approximately between the <2.555 Ga Glenburgh and the 3.7-3.0 Ga Narryer terranes.

The authors provide two alternative interpretations for the reflector. According to the first model, it would represent a frozen-in remnant of Archean subduction zone (preferred by the authors), and according to the second model, it would be a suture from thrusting of the Glenburgh terrane beneath the Yilgarn craton in the Proterozoic.

I have the following notes and comments on the manuscript.

(1)The manuscript lacks a good introductory paragraph pointing out what are the implications of the study beyond the Yilgarn craton evolution. Are we just reading a report on seismic reflectors in Yilgarn, or do we see a thorough discussion on the first plate tectonic subduction process? This should be

improved.

We recognise that we did not fully develop the implications of our work in the initial submission, and have now added new first and final paragraphs to place our work more clearly in the broader context of constraining the initiation of subduction during the Archean. This paper pushes seismic reflection evidence for subduction back in time from the Neoproterozoic to the end of the Mesoproterozoic, up to 130 Ma earlier than previous work.

(2) The authors seem to assume that Archean and Paleoproterozoic plate tectonic processes were similar to the present ones. Thus, they simply follow the uniformitarian principle, which is not actually correct when interpreting Archean processes. The geothermal conditions were quite different in Archean and Paleoproterozoic times due to much higher radiogenic heat production, higher mantle heat flow, and higher geothermal gradient. The tectonic mechanism must have operated differently at least until end of early Archean. The plates were much thinner, and it is still an open question when did plate tectonics start to operate in the modern sense, although ~3 Ga is often mentioned. Due to the hot lithosphere in the continental and oceanic areas, subduction may have not been possible at all until the earth had cooled sufficiently. As we know, subduction requires a positive density contrast between the slab and the upper mantle, and such a contrast may not have existed if the oceanic crust is assumed to have been basaltic. On the other hand, subduction may have started to operate early if the oceanic crust comprised high Mg-basalts (i.e., komatiites) due to high potential temperature of the mantle. Therefore, the authors have here an opportunity to indirectly imply something about the geothermal conditions and plate tectonic processes during the Archean.

In our new introductory paragraph we summarise possible early tectonic processes on Earth and their secular variation, noting how plate convergence can produce different structural styles as mantle temperatures decrease. In the space available, we cannot review all these issues, but we briefly explain how seismic reflection surveys might discriminate between subduction and pre-subduction processes by mapping the geometry of preserved structures in the lower crust and upper mantle.

(3) The interpretation of the Yilgarn mantle reflector as a suture zone is naturally the most plausible interpretation, especially when the geological evidence is considered, and similar seismic structures have been presented in literature for both Archean and Proterozoic areas with similar interpretations. However, taking into account the hot geothermal conditions, we should also consider different mechanisms, which would have produced similar structures. The geological character of the mantle reflector remains speculative, of course, but most probably it is due to eclogitized basaltic rock or shearing. If the terranes in Yilgarn had eclogitic lower crust already in the Archean (?) it would have allowed removal of the lowermost eclogitic crust by delamination in the orogenic peak temperatures, either as a slab or as a fluid, i.e. Rayleigh-Taylor-style flow over geologically short times scales (~10-30 Ma). Lower crustal mafic rock would have turned into eclogite at early stages of crustal thickening in collisions of arc-type crusts and plates. In the present conditions Moho is at >50 km under the Glenburgh terrane, and taking into account the layers removed by erosion (10-15 km?) it may have been much deeper initially.

To create the volume of felsic-intermediate crust found in Archean cratons, a large volume (>100 km total thickness?) of mafic/eclogitic restite must also have been generated, and it likely foundered into the underlying mantle, possibly on several occasions. We now mention that delamination could possibly produce the mantle reflections observed, but we note that delamination would not produce the hydrous volcanic rocks that are observed at the surface, and that we suggest are linked to the mantle reflections we observe.

We, however, mention that eclogitization may have played a role in modification of the crustal root created by underthrusting of the Glenburgh Terrane, and could be responsible for the change in lower crustal reflectivity we observe in this region.

(4) There are stack sections combining and comparing survey lines CP3 and YU1 (Fig. 2) and SC1 and YU1 (Suppl. Fig. 2). I would be keen to see also a figure showing SC1 and CP3. This would help in 3D visualization of the reflector.

We have now added an additional supplementary Fig. S4 to show the tie between the stacks of line SC1 and CP3.

Aug. 20, 2021
Ilmo Kukkonen

REVIEWERS' COMMENTS

Reviewer #2 (Remarks to the Author):

The authors have addressed my comments and it is much clearer the difference between the earlier studies and this current one both in terms of interpretations and re-processing work. I have no further comments.

Reviewer #3 (Remarks to the Author):

I'm happy with the revised manuscript.

Response to Reviewers

We sincerely appreciate the time and effort that the reviewers handling our paper, and conclude that only editorial changes to our manuscript are now required .

Reviewer #2 (Remarks to the Author):

The authors have addressed my comments and it is much clearer the difference between the earlier studies and this current one both in terms of interpretations and re-processing work. I have no further comments.

Reviewer #3 (Remarks to the Author):

I'm happy with the revised manuscript.